# Immune Checkpoint Inhibitors and Pregnancy: Analysis of the VigiBase^®^ Spontaneous Reporting System

**DOI:** 10.3390/cancers15010173

**Published:** 2022-12-28

**Authors:** Roberta Noseda, Laura Müller, Francesca Bedussi, Michele Fusaroli, Emanuel Raschi, Alessandro Ceschi

**Affiliations:** 1Division of Clinical Pharmacology and Toxicology, Institute of Pharmacological Sciences of Southern Switzerland, Ente Ospedaliero Cantonale, 6900 Lugano, Switzerland; 2Pharmacology Unit, Department of Medical and Surgical Sciences, Alma Mater Studiorum-University of Bologna, 40126 Bologna, Italy; 3Clinical Trial Unit, Ente Ospedaliero Cantonale, 6900 Lugano, Switzerland; 4Faculty of Biomedical Sciences, Università della Svizzera Italiana, 6900 Lugano, Switzerland; 5Department of Clinical Pharmacology and Toxicology, University Hospital Zurich, 8091 Zurich, Switzerland

**Keywords:** immune checkpoint inhibitors, pregnancy, safety, pharmacovigilance, VigiBase^®^, disproportionality analysis

## Abstract

**Simple Summary:**

In preclinical studies, it has been shown that the blockade of immune checkpoint pathways increases the risk of fetal death. Therefore, the use of immune checkpoint inhibitors (ICIs) in cancer patients who are either pregnant or of childbearing potential is not recommended. Nevertheless, some clinical cases have been published showing positive pregnancy-related outcomes. To characterize the ICI safety profile in pregnancy, we used VigiBase^®^, the World Health Organization’s spontaneous reporting system, and described 103 safety reports referring to ICI exposure during the peri-pregnancy period. Of these, 56 reported pregnancy-related outcomes, including spontaneous abortion, fetal growth restriction, and prematurity, for which we did not find signals of disproportionate reporting. Considering the expanding indications of ICIs, continuous surveillance by clinicians and pharmacovigilance experts is warranted.

**Abstract:**

In pregnancy, immune checkpoint pathways are involved in the maintenance of fetomaternal immune tolerance. Preclinical studies have shown that immune checkpoint inhibitors (ICIs) increase the risk of fetal death. Despite the fact that using ICIs in pregnant women and women of childbearing potential is not recommended, some case reports of ICI exposure in pregnancy have been published showing favorable fetal outcomes. This study aimed to gain further insight into ICI safety in pregnancy by querying VigiBase^®^, the World Health Organization’s spontaneous reporting system. We performed raw and subgroup disproportionality analyses using the reporting odds ratio and comparing ICIs with the entire database, other antineoplastic agents, and other antineoplastic agents gathered in VigiBase^®^ since 2011. Across 103 safety reports referring to ICI exposure during the peri-pregnancy period, 56 reported pregnancy-related outcomes, of which 46 were without concomitant drugs as potential confounding factors. No signals of disproportionate reporting were found for spontaneous abortion, fetal growth restriction, and prematurity. In light of the expanding indications of ICIs, continuous surveillance by clinicians and pharmacovigilance experts is warranted, along with pharmacoepidemiological studies on other sources of real-world evidence, such as birth records, to precisely assess ICI exposure during the peri-pregnancy period and further characterize relevant outcomes.

## 1. Introduction

With the expanding use of immune checkpoint inhibitors (ICIs) across cancer types and their potential for long-term disease control, their use in pregnancy may become increasingly common [1]. Immune checkpoint pathways involving the programmed cell death-1 protein (PD-1), its ligands (programmed cell death-ligand 1 and 2, PD-L1 and PD-L2), and the cytotoxic T-lymphocyte-associated protein-4 (CTLA-4) play a physiological role in maintaining maternal immune tolerance to the developing fetus [2,3]. It is noteworthy that the PD-L1 expression on T cells at the maternal–fetal interface increases during the progression of pregnancy to prevent fetal in-utero rejection [4,5,6]. Therefore, by blocking these pathways, ICIs could theoretically provoke an immune response against the fetus [7]. Moreover, as immunoglobulin G antibodies, ICIs can cross the placental barrier with potential transmission from the mother to the developing fetus, potentially resulting in an increased risk of immune-mediated disorders [8,9]. Preclinical studies in murine models and cynomolgus monkeys have shown that the PD-1/PD-L1 blockade in pregnancy disrupts tolerance to the developing fetus. Even if neither malformations nor immunological complications are observed in the offspring [4,5], this dysregulation may result in fetal death [10]. Therefore, using ICIs in pregnant women and women of childbearing potential (in the lack of effective contraception) is not recommended unless the clinical benefit outweighs the potential risk [11]. Nevertheless, some case reports on ICI exposure before, at, or after conception have shown favorable fetal outcomes without developmental abnormalities [12,13,14,15,16,17,18,19,20,21,22,23].

Considering the inherent challenges and limitations of pregnancy research, including ethical issues for inclusion in clinical trials, large-scale spontaneous reporting systems represent a privileged source of real-world data to investigate pregnancy-related adverse events. Therefore, to gain further insight into ICI safety in pregnancy, we queried VigiBase^®^, the World Health Organization’s (WHO) global pharmacovigilance database of safety reports of suspected adverse drug reactions (ADRs) [24]. Herein, we described the largest-to-date series of cases from clinical practice referring to ICI exposure during the peri-pregnancy period and the possible reporting of pregnancy-related outcomes.

## 2. Materials and Methods

### 2.1. Study Design

We performed a disproportionality analysis sided by a case-by-case evaluation on VigiBase^®^ de-duplicated safety reports gathered in the database from its inception through to 30 April 2022. Drugs of interest were ICIs. Events of interest were pregnancy-related outcomes. This complementary design was implemented to provide an exhaustive pharmacovigilance perspective.

### 2.2. Data Source

VigiBase^®^ is the global pharmacovigilance database developed and maintained by the WHO-Uppsala Monitoring Centre (WHO-UMC). It collects over 30 million safety reports of suspected ADRs from the national centers for pharmacovigilance participating in the WHO Programme for International Drug Monitoring (WHO PIDM). Member countries of the WHO PIDM (more than 170 in 2022) can, in turn, access VigiBase^®^ and analyze its content. The purpose of VigiBase^®^ is to identify unexpected ADRs. Moreover, it can be used as a reference source to gain knowledge on the safety profile of specific drugs and/or special populations such as pregnant women [24,25,26,27,28,29,30].

### 2.3. Selection Criteria of Safety Reports

We retrieved ICI exposure by searching, among the suspected drugs, for the following active ingredients: ipilimumab, nivolumab, pembrolizumab, cemiplimab, atezolizumab, avelumab, durvalumab, and dostarlimab. We retrieved pregnancy-related outcomes using the standardized query “pregnancy and neonatal topics”, available through the medical dictionary for regulatory activities (MedDRA^®^, version 25.0), on which VigiBase^®^ relies for the codification of reported ADRs. To further restrict to safety reports of ICI exposure during the peri-pregnancy period, we searched for any of the following events: “exposure via body fluid”; “exposure via father”; “fetal exposure during pregnancy”; “maternal exposure before pregnancy”; “maternal exposure during breastfeeding”; “maternal exposure during pregnancy”; “maternal exposure timing unspecified”; “paternal drugs affecting fetus”; “paternal exposure before pregnancy”; “paternal exposure timing unspecified”; “pregnancy”; “pregnancy with contraceptive device”. To be sure to include all safety reports of ICI exposure during the peri-pregnancy period in the study cohort, we also manually revised safety reports captured by the standardized query “pregnancy and neonatal topics” that, although not reporting events specifying ICI exposure during the peri-pregnancy period, reported specific pregnancy-related outcomes. 

### 2.4. Variables

The demographic and clinical characteristics of the included safety reports were described, including general information (country of origin, reporting year, and type of reporter), patient data (sex and age at ADR onset), and ICI treatment (time of exposure in relation to pregnancy, regimen, and indication). Pregnancy-related outcomes were classified into maternal and fetal/neonatal. Maternal events were further divided into specific pregnancy complications and more general ones. Concomitant drugs were assessed as potential confounding factors for reporting pregnancy-related outcomes by searching in the Reprotox^®^ database (https://www.reprotox.org/, last access on 25 October 2022).

### 2.5. Disproportionality Analyses

By gathering millions of safety reports, VigiBase^®^ allows data mining to identify potential safety signals. Disproportionality analyses (also known as the case/non-case approach) detect adverse events more often reported in individuals exposed to ICIs than in individuals exposed to other drugs [31].

To reduce the likelihood of false positives, we performed disproportionality analyses for pregnancy-related outcomes reported in at least five of the included ICI-related safety reports [32]. We used the reporting odds ratio (ROR) and considered it to be significant when the lower limit of its 95% confidence interval was >1 [32]. We used three reference groups as comparators: (i) the entire database, (ii) only safety reports suspecting antineoplastic agents different from ICIs (using the anatomical, therapeutic and chemical, ATC, code L01) to control confounding by indication [33], and (iii) only safety reports suspecting antineoplastic agents different from ICIs and submitted after 2011 (when the first-in-class ipilimumab received marketing authorization by the Food and Drug Administration). Subgroup disproportionality analyses perform better than raw ones in terms of sensitivity and precision in large databases such as VigiBase^®^ [34,35]. Moreover, subgroup disproportionality analyses may uncover susceptibilities to ADRs in VigiBase^®^ [35]. Therefore, after performing raw disproportionality analyses, we repeated the same analyses on subgroups of safety reports concerning women aged 20–44 years (based on the demographic characteristics concerning patient sex and age of the safety reports included in the study). 

Data management and analyses were performed with Microsoft Excel (2010, Microsoft Corporation, Washington, DC, USA) and GraphPad Prism 9 (GraphPad Software Inc., San Diego, CA, USA).

According to the Human Research Act (810.30, of 30 September 2011—status as of 1 December 2022), from the Federal Assembly of the Swiss Confederation, ethical approval and written informed consent were not required (Art. 2: “It does not apply to research which involves anonymously collected or anonymized health-related data”).

## 3. Results

As of 30 April 2022, 30,438,983 de-duplicated safety reports were gathered in VigiBase^®^, including 123,289 safety reports with ICIs as suspected drugs. Of these, 615 reported events featured in the SMQ “pregnancy and neonatal topics”. After excluding 512 safety reports that did not meet the predefined inclusion criteria, 103 safety reports were included in the study cohort (Figure 1): 100 safety reports reporting events specifying ICI exposure during the peri-pregnancy period and 3 safety reports that, although lacking the latter events, reported specific pregnancy-related outcomes (i.e., spontaneous abortion in two cases and neonatal respiratory distress syndrome in one case).

### 3.1. Demographic and Clinical Characteristics of Safety Reports

Table 1 shows the demographic and clinical characteristics of the safety reports included in the study. The majority of safety reports came from the United States of America (65, 63.1%) and reporting peaked in 2019 with 25 (24.3%) safety reports, and then declined in subsequent years. In 86 (83.5%) safety reports, the reporter was a healthcare professional. The median age was 32 years (ranging from 20 to 44 years, interquartile range 28–35 years, *n* = 45), with 96 (93.2%) cases described in women. Maternal exposure to ICIs occurred during pregnancy in 77 (74.8%) safety reports. Only in 3 (2.9%) safety reports were women treated with ICIs before pregnancy. A 43-year-old woman affected by malignant melanoma reported an abortion induced three months after the end of treatment with pembrolizumab. Two other women (39 years old, Hodgkin’s lymphoma, and 38 years old, malignant melanoma) reported spontaneous abortion about one year and two years after the end of anti-PD-1 monotherapy with nivolumab and pembrolizumab, respectively. The PD-1/PD-L1 pathway was the target of ICI treatment in 76 (73.8%) safety reports and malignant melanoma was the underlying cancer type in 28 (27.2%) safety reports.

### 3.2. Characterization of Pregnancy-Related Outcomes

Out of 103 safety reports, 47 (45.6%) reported only exposure to ICIs during the peri-pregnancy period, while 56 (54.4%) also reported 104 pregnancy-related outcomes (more than one outcome was recorded in some safety reports) (Appendix A). Of these, 36 were maternal and 68 fetal/neonatal (Table 2). Specific maternal pregnancy complications occurred in three cases and included pre-eclampsia, HELLP syndrome (hemolysis, elevated liver enzymes, and low platelet count) with a placental disorder, and a case of placental infarction. Among 32 more general maternal outcomes, no specific toxicity patterns were observed. Regarding fetal/neonatal pregnancy-related outcomes, five safety reports reported normal newborn/live birth, while fatal events occurred in two safety reports. No patterns of major birth defects and no patterns of specific immune-related adverse events were found.

Out of 56 safety reports with pregnancy-related outcomes, 10 (17.9%) had a median of 2 (IQR 1–4) concomitant drugs (Appendix A). However, in six cases, concomitant drugs were likely administered to stimulate fetal lung maturation (betamethasone in five cases and dexamethasone in one case [36]), to premature neonates for pain management (morphine and oxycodone in one case [37,38]), and as prophylactic antifungal treatment of the premature neonate (in one case [39]). One case of umbilical cord compression and hypoxia reported four antineoplastic agents as concomitant drugs for which the assessment as potential confounding factors was not applicable. Lastly, insulin and ramipril were reported as concomitant drugs in a case of a successful normal newborn. 

### 3.3. Disproportionality Analyses

We performed raw disproportionality analyses for spontaneous abortion, fetal growth restriction, and prematurity with at least five safety reports. We found no signal of disproportionate reporting in any of the three predefined comparator groups (Table 3 and Figure 2).

For spontaneous abortion and prematurity, we performed the subgroup analysis in females aged 20–44 years (no safety reports fulfilled the criteria for fetal growth restriction). Again, we found no signal of increased reporting with any of the three predefined comparator groups (Table 3 and Figure 2).

## 4. Discussion

This pharmacovigilance study in VigiBase^®^ provided the largest-to-date series of cases referring to ICI exposure during the peri-pregnancy period and reporting pregnancy-related outcomes. We found no specific patterns of maternal, fetal, or neonatal toxicity. No signal of disproportionate reporting was detected for spontaneous abortion, fetal growth restriction, or prematurity with ICIs, although 104 pregnancy-related outcomes were identified from 56 patients.

Notwithstanding the expanding use of ICIs across multiple indications and settings [1], current knowledge about ICI exposure during the peri-pregnancy period consists of preclinical data and some clinical cases from the scientific literature, with apparently contrasting evidence. On the one hand, studies in animal models with anti-PD-1/PD-L1 agents showed an increased risk of pregnancy loss [4,5,6,7,10], plausibly related to the role played by immune checkpoints in maintaining fetomaternal immune tolerance [2,3]. On the other hand, published case reports of women who were receiving ICI treatment at the time of conception [16,18], or who started ICI treatment during pregnancy [12,13,14,15,17,18,19,21,23], or who became pregnant after the end of ICI treatment [22] overall reported favorable pregnancy outcomes. In contrast, a recent systematic review of case reports also performed a search in the Food and Drug Administration Adverse Event Reporting System (FAERS) and found different pregnancy complications associated with ICIs, including intrauterine growth restriction, spontaneous abortion, premature delivery, and fetal distress syndrome [20]. These findings suggest that published case reports/series could suffer from positive-result bias [40], thus leaving open doubts about ICI safety in pregnancy.

In the study cohort, the more common reason for using ICIs was melanoma, which is also the most common malignancy diagnosed during pregnancy [41]. Similar to the published clinical cases [12,13,14,15,17,18,19,21,23], we observed that in most of the safety reports, ICI exposure occurred during pregnancy, whereas no reports indicated ICI treatment at the time of conception. It is noteworthy that three safety reports recorded pregnancy-related outcomes after ICI treatment cessation with a temporal gap ranging from three months to two years. According to current recommendations that advise patients of childbearing age to avoid ICI exposure (unless using effective contraception) during and for at least 5 months after the last dose of ICI treatment [3], the woman reporting an induced abortion 3 months after pembrolizumab discontinuation may still have been exposed to the drug. In terms of efficacy, durable responses with ICIs are becoming increasingly common, suggesting that ICIs could have long-term physiological implications through molecular mechanisms that are still not entirely clear [42,43]. Indeed, the extended duration of their therapeutic effects does not match with their pharmacokinetic half-lives, with an increased risk of late toxicity not only during prolonged active treatments, but also after treatment discontinuation (albeit more sporadically) [43,44,45,46].

With regard to pregnancy, immune checkpoint pathways are involved in the establishment and maintenance of maternal immune tolerance to the semi-allogeneic fetus [9]. Therefore, treatment with ICIs could theoretically negatively affect the immune processes underpinning the fetomaternal immune tolerance during pregnancy even months (or years) after the end of ICI treatment. Moreover, besides this direct effect of ICIs on the maternal immune system at the placental barrier, as monoclonal antibodies, ICIs undergo active transport across the placental barrier through the neonatal Fc receptors [47]. These, albeit absent during organogenesis (up to the fourteenth gestational week), increase in the late second and the third trimesters [8,10]. It is noteworthy that individual receptor occupancy might depend on several factors, including tumor burden and genetic polymorphisms, affecting the neonatal Fc receptors [3]. Therefore, because of such an individual variability, long-term toxicities of ICIs during the peri-pregnancy period might occur within a timeframe that cannot be unambiguously defined, thus providing a rationale for the assessment of ICI safety in cases whereby ICI exposure was reported up to two years prior to pregnancy.

In preclinical studies, the risk of spontaneous abortion was increased five-fold in models of allogeneic mice pregnancy treated with pharmacological inhibition of PD-L1, but not in syngeneic ones [4], suggesting that fetomaternal immune tolerance and PD-L1 expression at the utero–placental interface could be modulated by the degree of fetal allogeneity [4,7,10]. Therefore, the effects of anti-PD-1/PD-L1 antibodies on the fetus are anticipated to be patient-specific and strongly linked to the paternal antigenic components [43]. Another role whereby the father might be involved is as an oncologic patient treated with ICIs. To date, it is unknown whether paternal treatment with ICIs might affect pregnancy. Nevertheless, it is worth mentioning that in our case series, a few safety reports of paternal ICI exposure were included. Remarkably, in one case, the father was treated with ICIs before conception (although it was unknown how long before ICI treatment was interrupted), whereas in the remnants, the time of exposure was not documented.

Most of the safety reports included in the present study were from healthcare professionals, suggesting a spreading awareness about the potential negative effects of ICI use in pregnant women or those of childbearing age. Moreover, slightly less than half of the safety reports referred to ICI exposure during the peri-pregnancy period without mentioning any type of pregnancy-related outcomes. 

This finding might further support the fact that among healthcare professionals, the use of ICIs during the peri-pregnancy period, whenever chosen, remains suspicious and prompts them to spontaneously report their use not in accordance with product labels, even in the absence of pregnancy complications.

Interestingly, we observed that the reporting of safety reports concerning ICI exposure during the peri-pregnancy period peaked in 2019 and then declined over the following two years. This observation might suggest that, despite the widespread awareness mentioned above, the knowledge provided in the last two or three years by clinical cases describing positive pregnancy outcomes could have influenced the attitude of reporters, who, in the absence of pregnancy complications, may have stopped reporting ICI exposure alone. In addition, the decline observed from 2019 onwards could be related to the fact that during the COVID-19 pandemic, in several healthcare systems, a delay of several cancer diagnoses with the postponement of related therapies occurred.

With regard to pregnancy-related outcomes, neither maternal specific pregnancy complications nor patterns of immune-related adverse events were observed, in line with the published single clinical cases [12,13,14,15,16,17,18,19,20,21,22,23]. Noteworthy, one case of maternal hypophysitis occurred, which is a well-described endocrine toxicity with ICIs [48] with potential negative consequences on gonadal function and fertility [9]. Concerning fetal/neonatal outcomes, few safety reports reported successful pregnancy outcomes, and no patterns of major birth defects and no specific immune-related adverse events were found.

### Strengths and Limitations

As VigiBase^®^ is the largest spontaneous reporting system collecting safety reports worldwide, disproportionality analyses performed in this database support the generalizability of results. While disproportionality analyses are established approaches for signal detection of rare ADRs, a few studies have specifically addressed pregnancy-related outcomes [25,26,27,28,29,30]. In this setting, which concerns a niche of safety reports, we carried out a rigorous case selection and assessment, accounting for major biases (e.g., confounding by indication and concomitant drugs). Nonetheless, we acknowledged major drawbacks of spontaneous reporting systems, including over- and under-reporting (which also limits the sensitivity of signal detection by disproportionality analysis [31]), partial and missing information, and unavailability of clinical details (in the specific setting of ICIs, concerning, e.g., cancer stage, duration of ICI treatment, patient comorbidities, and exact trimester of ICI exposure). Moreover, VigiBase^®^ does not allow for firmly inferring causality, as safety reports are based on the reporter’s suspicion of a causal relationship between drug use and the onset of adverse events, without information on differential diagnoses. Lastly, follow-up information on children is not collected.

Furthermore, disproportionality analysis per se has some limitations. The ROR is a statistical estimate that does not inform about the real risk of developing a certain ADR, but only indicates an increased risk of reporting that ADR. As the denominator (i.e., the exposed population) is unknown, ROR computation does not inform about the incidence of ADRs. Therefore, disproportionality analysis can only generate hypotheses that eventually need to be further investigated.

## 5. Conclusions

By exploiting VigiBase^®^, we evaluated the largest series of cases referring to ICI exposure during the peri-pregnancy period and reporting of pregnancy-related outcomes: no signal of disproportionate reporting was detected for spontaneous abortion, fetal growth restriction, and prematurity with ICIs, although pregnancy-related outcomes were found in 56 safety reports. 

Considering the expanding uses of ICIs in general and in the subpopulation of pregnant women or those of childbearing age and wishing to become pregnant who are suffering from cancer, we support continuous surveillance by clinicians and pharmacovigilance experts of large-scale spontaneous reporting systems. As disproportionality analyses rely on the number of safety reports gathered in the spontaneous reporting system, the results from these analyses for the pregnancy-related outcome(s) of interest might change over time, thus making the reassessment of the ICI safety profile in the peri-pregnancy period of the utmost importance. Moreover, pharmacoepidemiological studies on different sources of real-world evidence, such as birth records, are warranted to precisely assess ICI exposure during the peri-pregnancy period and to further characterize relevant outcomes.

## Figures and Tables

**Figure 1 cancers-15-00173-f001:**
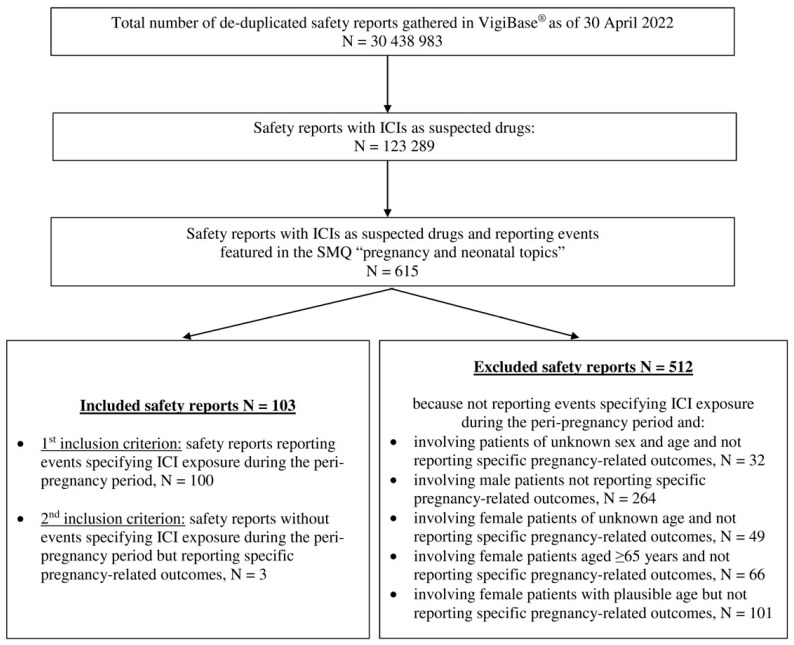
Flowchart. ICI, immune checkpoint inhibitor; SMQ, standardized MedDRA query.

**Figure 2 cancers-15-00173-f002:**
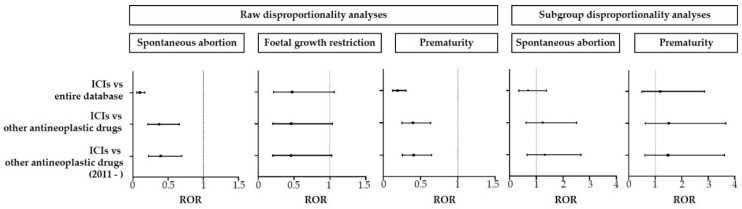
Raw and subgroup (by sex and age, females aged 20–44 years) disproportionality analyses between immune checkpoint inhibitors and the entire database, other antineoplastic agents, and other antineoplastic agents since 2011, to compare the reporting of spontaneous abortion, fetal growth restriction, and prematurity. Forest plots represent reporting odds ratios along with 95% confidence intervals. ICI, immune checkpoint inhibitor; ROR, reporting odds ratio; CI, confidence interval.

**Table 1 cancers-15-00173-t001:** Demographic and clinical characteristics of the safety reports included in the study.

Characteristic	*n* (%), *N* = 103
Country of origin	
United States of America	65 (63.1)
Europe	30 (29.1)
South America	4 (3.9)
Australia	4 (3.9)
Reporting year	
2012	1 (1.0)
2014	1 (1.0)
2015	3 (2.9)
2016	12 (11.7)
2017	13 (12.6)
2018	13 (12.6)
2019	25 (24.3)
2020	18 (17.5)
2021	12 (11.7)
2022 (as of 30 April)	5 (4.9)
Type of reporter	
Physician	52 (50.5)
Other healthcare professional	25 (24.3)
Pharmacist	9 (8.7)
Consumer	15 (14.6)
Not reported	2 (1.9)
Patient sex	
Female	96 (93.2)
Male ^1^	6 (5.8)
Not reported ^2^	1 (1.0)
Patient age	
Reported	50 (48.5)
In neonates (<30 days)	5
In adults	45
Not reported	53 (51.5)
Time of ICI exposure	
Maternal exposure during pregnancy ^3^	77 (74.8)
Exposure via father ^4^	12 (11.7)
Maternal exposure timing unspecified	11 (10.7)
Maternal exposure before pregnancy	3 (2.9)
ICI regimen	
Anti-CTLA-4 monotherapy	
ipilimumab	12 (11.7)
Anti-PD-1 monotherapy	
nivolumab	42 (40.8)
pembrolizumab	29 (28.2)
Anti-PD-L1 monotherapy	
atezolizumab	5 (4.9)
Combination of	
nivolumab and ipilimumab	7 (6.8)
nivolumab and ipilimumab in regimen not definable ^5^	8 (7.8)
Indication	
Malignant melanoma	28 (27.2)
Hodgkin’s lymphoma	12 (11.7)
Renal cell carcinoma	7 (6.8)
Colon cancer	3 (2.9)
Lung cancer	3 (2.9)
Lymphoma	2 (1.9)
Glioma	1 (1.0)
Alveolar soft part sarcoma	1 (1.0)
Pericardial mesothelioma	1 (1.0)
Breast cancer	1 (1.0)
Gestational throphoblastic tumor	1 (1.0)
Not reported	43 (41.7)

^1^ One safety report referred to paternal exposure to nivolumab and reported neonatal outcomes; six safety reports involved male neonates. ^2^ Safety report involving a neonate of unknown sex. ^3^ One safety report reported maternal exposure to pembrolizumab both during pregnancy and during breastfeeding. ^4^ One safety report of paternal exposure before pregnancy (unknown how long before) and eleven safety reports with exposure via father at an unspecified time of pregnancy. ^5^ Because of partially recorded or missing dates of administration. IQR, interquartile range; ICI, immune checkpoint inhibitor; CTLA-4, cytotoxic T-lymphocyte antigen-4; PD-1, programmed cell death-1; PD-L1, programmed cell death-ligand 1.

**Table 2 cancers-15-00173-t002:** Pregnancy-related outcomes reported on immune checkpoint inhibitors in VigiBase^®^ as of 30 April 2022.

Pregnancy-Related Outcomes.	*n* (%), *N* = 56 *
Maternal outcomes	
Specific pregnancy complications	
Pre-eclampsia	1 (1.8)
HELLP syndrome	1 (1.8)
Placental disorder	1 (1.8)
Placental infarction	1 (1.8)
More general outcomes	
Diarrhea	3 (5.4)
Nausea	2 (3.6)
Fatigue	2 (3.6)
Abdominal pain	2 (3.6)
Pruritus	2 (3.6)
Chest pain	2 (3.6)
Diabetes mellitus	1 (1.8)
Hypophysitis	1 (1.8)
Arthralgia	1 (1.8)
Hypophagia	1 (1.8)
Starvation	1 (1.8)
Ketoacidosis	1 (1.8)
Urinary tract infection	1 (1.8)
Neutropenia	1 (1.8)
Lung disorder	1 (1.8)
Iron deficiency anemia	1 (1.8)
Antiphospholipid syndrome	1 (1.8)
Abdominal distension	1 (1.8)
Autoimmune disorder	1 (1.8)
Anxiety	1 (1.8)
Cardiac disorder	1 (1.8)
Tri-iodothyronine increased	1 (1.8)
Insomnia	1 (1.8)
Dyspnea	1 (1.8)
Breastfeeding	1 (1.8)
Fetal/neonatal outcomes	
Normal newborn	4 (7.1)
Live birth	1 (1.8)
Fetal death	1 (1.8)
Stillbirth	1 (1.8)
Spontaneous abortion	12 (21.4)
Abortion induced	7 (12.5)
Spontaneous abortion incomplete	1 (1.8)
Fetal growth restriction	6 (10.7)
Fetal distress syndrome	1 (1.8)
Small for gestational age	1 (1.8)
Umbilical cord compression	1 (1.8)
Prematurity	18 (32.1)
Neonatal respiratory distress syndrome	2 (3.6)
Hypoxia	1 (1.8)
Lung disorder	1 (1.8)
C-reactive protein increased	1 (1.8)
White blood cell count increased	1 (1.8)
Retinopathy of prematurity	1 (1.8)
Intraventricular haemorrhage neonatal	1 (1.8)
Motor developmental delay	1 (1.8)
Neonatal type 1 diabetes mellitus	1 (1.8)
Birth defects	
Congenital hand malformation	1 (1.8)
Congenital pulmonary valve disorder	1 (1.8)
Congenital hypothyroidism	1 (1.8)
Hypospadias	1 (1.8)

* Some safety reports reported multiple pregnancy-related outcomes.

**Table 3 cancers-15-00173-t003:** Computation of reporting odds ratios and 95% confidence intervals in raw and subgroup (by sex and age, females aged 20–44 years) disproportionality analyses.

		Raw Disproportionality Analyses					Subgroup Disproportionality Analyses				
		(a)	(b)	(c)	(d)	ROR [95% CI]	(a)	(b)	(c)	(d)	ROR [95% CI]
Spontaneous abortion	ICIs vs. entire database	12	123,277	30,612	30,408,371	0.097 [0.055–0.170]	8	2802	16,805	4,082,848	0.694 [0.346–1.389]
	ICIs vs. other antineoplastic agents	12	123,277	715	2,727,718	0.371 [0.210–0.657]	8	2802	367	159,606	1.242 [0.616–2.504]
	ICIs vs. other antineoplastic agents (2011–)	12	123,190	593	2,401,037	0.394 [0.223–0.698]	8	2792	298	137,342	1.321 [0.654–2.668]
Fetal growth restriction	ICIs vs. entire database	6	123,283	3117	30,435,866	0.475 [0.213–1.059]	-	2810	604	4,099,049	NC
	ICIs vs. other antineoplastic agents	6	123,283	288	2,728,145	0.461 [0.205–1.035]	-	2810	39	159,934	NC
	ICIs vs. other antineoplastic agents (2011–)	6	123,196	256	2,401,374	0.457 [0.203–1.026]	-	2800	32	137,608	NC
Prematurity	ICIs vs. entire database	18	123,271	22,998	30,415,985	0.193 [0.122–0.307]	5	2805	6128	4,093,525	1.191 [0.495–2.864]
	ICIs vs. other antineoplastic agents	18	123,271	1000	2,727,433	0.398 [0.250–0.635]	5	2805	189	159,784	1.507 [0.620–3.665]
	ICIs vs. other antineoplastic agents (2011–)	18	123,184	861	2,400,769	0.407 [0.255–0.650]	5	2795	166	137,474	1.481 [0.608–3.609]

(a) Number of safety reports concerning the pregnancy-related outcome of interest reported in association with ICIs. (b) Number of safety reports concerning outcomes other than the one of interest reported in association with ICIs. (c) Number of safety reports concerning the pregnancy-related outcome of interest reported in association with all other drugs present in the entire database, other antineoplastic agents, and other antineoplastic agents (since 2011), respectively. (d) Number of safety reports concerning outcomes other than the one of interest reported in association with all other drugs present in the entire database, other antineoplastic agents, and other antineoplastic agents (since 2011), respectively. ICI, immune checkpoint inhibitor; ROR, reporting odds ratio; CI, confidence interval; NC, not calculated (because there were no safety reports of fetal growth restriction fulfilling the criteria used to define the subgroups).

## Data Availability

The data presented in this study are openly available from www.vigiaccess.org (accessed on 30 April 2022) by applying the criteria described in the Methods section of the manuscript.

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
