# Peer review of "Immune Checkpoint Inhibitors and Pregnancy: Analysis of the VigiBase® Spontaneous Reporting System"

_cancers, 2022, doi:10.3390/cancers15010173_

Round 1

Reviewer 1 Report

Please better clarify in table 1 what "median [IQR], range in years"  means. probably you should separate this row from others reporting the number of report, since they regard a different variable with a different unit of measure. 

Reviewer 2 Report

The paper entitled “Immune checkpoint inhibitors and pregnancy: Analysis of the VigiBase® spontaneous reporting system” aimed to gain further insight into ICI safety in pregnancy by querying VigiBase®, the World Health Organization’s spontaneous reporting system. After analyzing 103 safety reports referring to ICI exposure during the peri-pregnancy period, the authors found no signals of disproportionate reports for spontaneous abortion, fetal growth restriction, and prematurity. In summary, this manuscript addressed a central topic of well biomedical and clinical importance. However, some descriptions about the Tables and Figures are not informative enough and the Discussion part is not exhaustive. The following concerns should be addressed to improve the quality of this manuscript.

1. In Line 140, the authors claimed that “As of 30 April 2022, 385289 safety reports reported events featured in the SMQ ‘pregnancy and neonatal topics’”. This information was not presented in Figure 1. In addition, the contents in the first two boxes of Figure 1 were not described in text, which may make the readers confuse about the samples included in this study.

2. The format of Figure 1 should be improved. For example, the contents in the box describing “Excluded safety reports” are not presented properly. In addition, more arrows are needed in the left box.

3. In Table 1, some information of safety reports was not in detailed in the results part, such as “Country of origin”, “Reporting year”, and “ICI regimen”. Please add more description about these data in the results part.

4. In 3.2 part, the sentences relating to the data in Table 2 and Table S1-S2 were not informative enough. Please expand accordingly.

5. In Table 3 and Figure 2, much information was presented. However, in 3.3 part, the explanations and description were not exhaustive enough. The authors are suggested to provide more details.

6. In Discussion part, although the authors have provided many discussions about the similarity and difference between findings from current study and previous reports, the future applications and directions of this field were not presented.

7. Some sentences in “Simple Summary” and “Abstract” are repeated. Please rephrase these sentences.

Round 2

Reviewer 2 Report

The authors have addressed all my comments for this paper and the manuscript has been significantly improved. Therefore, I consider the paper is acceptable for publication.